# DeepI2I: Enabling Deep Hierarchical Image-to-Image Translation by Transferring from GANs

**Yaxing Wang, Lu Yu, Joost van de Weijer**
Computer Vision Center, Universitat Autònoma de Barcelona
{yaxing, lu, joost}@cvc.uab.es

## Abstract

Image-to-image translation has recently achieved remarkable results. But despite current success, it suffers from inferior performance when translations between classes require large shape changes. We attribute this to the high-resolution bottlenecks which are used by current state-of-the-art image-to-image methods. Therefore, in this work, we propose a novel deep hierarchical Image-to-Image Translation method, called *DeepI2I*. We learn a model by leveraging hierarchical features: (a) *structural information* contained in the shallow layers and (b) *semantic information* extracted from the deep layers. To enable the training of deep I2I models on small datasets, we propose a novel transfer learning method, that transfers knowledge from pre-trained GANs. Specifically, we leverage the discriminator of a pre-trained GANs (i.e. BigGAN or StyleGAN) to initialize both the encoder and the discriminator and the pre-trained generator to initialize the generator of our model. Applying knowledge transfer leads to an alignment problem between the encoder and generator. We introduce an *adaptor network* to address this. On many-class image-to-image translation on three datasets (Animal faces, Birds, and Foods) we decrease mFID by at least 35% when compared to the state-of-the-art. Furthermore, we qualitatively and quantitatively demonstrate that transfer learning significantly improves the performance of I2I systems, especially for small datasets. Finally, we are the first to perform I2I translations for domains with over 100 classes. Our code and models are made public at: `https://github.com/yaxingwang/DeepI2I`.

## 1 Introduction

Most image-to-image (I2I) networks have a high-resolution bottleneck [9, 21, 33, 34, 69, 36, 37, 67, 73, 74, 76]. These methods only apply two down-sampling blocks. Whereas such models are known to be successful for style transfer, it might be difficult to extract abstract semantic information. Therefore, we argue (and experimentally verify) that current I2I architectures have a limited capacity to translate between classes with significant shape changes (e.g. from dog face to meerkat face). To successfully translate between those domains a semantic understanding of the image is required. This information is typically extracted in the deep low-resolution layers of a network.

The loss of spatial resolution, which occurs when adding multiple down-sampling layers, is one of the factors which complicates the usage of deep networks for I2I. For high-quality I2I both low-level style information, as well as high-level semantic information needs to be transferred from the encoder to the generator of the I2I system. Therefore, we propose a deep hierarchical I2I translation framework (see Figure 1 right) that fuses feature representations at various levels of abstraction (depth). Leveraging the hierarchical framework, allows us to perform I2I translation between classes with significant shape changes. The proposed method, *DeepI2I*, builds upon the state-of-the-art BigGAN model [7], extending it to I2I translation by adding an encoder with the same architecture as the discriminator. An additional advantage of this architecture is that because it applies a latent class embedding, it

is scalable to *many-class domains*. We are the first to perform translations in a single architecture between over 100 classes, while current systems have only considered up to 40 classes maximum[1].

Another factor, that complicates the extension to deep I2I, is that the increasing amount of network parameters limits its applicability to I2I problems with large datasets. To address this problem we propose a novel knowledge transfer method for I2I translation. Transferring knowledge from pre-trained networks has greatly benefited the discriminative tasks [13], enabling the re-use of high-quality networks. However, it is not clear on what dataset a high-quality pre-trained I2I model should be trained, since translations between arbitrary classes might not make sense (e.g. translating from satellite images to dog faces does not make sense, while both classes have been used in separate I2I problems [22]). Instead, we propose to initialize the weights of the I2I model with those of a pre-trained GANs. We leverage the discriminator of a pre-trained BigGAN (Figure 1 left) to initialize both the encoder and the discriminator of I2I translation model, and the pre-trained generator to initialize the generator of our I2I model. To address the misalignment between the different layers of the encoder and generator, we propose a dedicated adaptor network to align both the pre-initialized encoder and generator.

We perform experiments on multiple datasets, including complex datasets which have significant shape changes such as animal faces [38] and foods [31]. We demonstrate the effects of the deep hierarchical I2I translation framework, providing qualitative and quantitative results. We further evaluate the knowledge transfer, which largely accelerates convergence and outperforms DeepI2I trained from scratch. Additionally, leveraging the adaptor is able to tackle the misalignment, consistently improving the performance of I2I translation.

## 2   Related work

**Generative adversarial networks.**   GANs are composed of a generator and a discriminator [18]. The generator aims to synthesize a fake data distribution to fool the discriminator, while the discriminator aims to distinguish the synthesized from the real distribution. Recent works [3, 19, 40, 42, 53, 72] focus on addressing the mode collapse and training instabilityproblems, which occur normally when training GANs. Besides, several works [7, 12, 27, 28, 50] explore constructing effective architectures to synthesize high-quality images. For example, Karras et al. [27, 28, 29] manage to generate progressively a high-realistic image from low to high resolution. BigGAN [7] successfully synthesizes high-fidelity images on imageNet [11].

**Image-to-image translation.**   Image-to-image translation has been studied widely in computer vision. It has achieved outstanding performance on both paired [17, 23, 77, 65, 64] and unpaired image translation [37, 33, 73, 76, 60, 5, 15, 61, 41, 49, 59]. These approaches, however, suffer from two main challenges: diversity and scalability. The goal of diversity translation is to synthesize multiple plausible outputs of the target domain from a single input image [21, 34, 2, 51, 74]. Scalability [9, 74, 10, 35] refers to translations across several domains using a single model. Several concurrent works [30, 71, 52, 56] also focus on *shape* translation as well as *style*. TransGaGa [71] firstly disentangles the input image in the geometry space and style space, then it conducts the translation for each latent space separately. However, none of these approaches addresses the problem of transfer learning for I2I.

**Transfer learning for GANs.**   A series of recent works investigated knowledge transfer on generative models [46, 70, 66, 54] as well as discriminative models [13, 48, 47, 62]. TransferGAN [70] indicates that training GANs on small data benefits from the pre-trained GANs. Noguchi et al. [46] study only updating part of the parameters of the pre-trained generator. Wang et al. [66] propose a minor network to explore the latent space of the pre-trained generator. Several other approaches [45, 44, 24, 16] study pre-trained generators, but do not focus on transferring knowledge. Furthermore, some methods [4, 1, 75] perform image editing by leveraging the pre-trained GAN. Image2StyleGAN [1] leverages an embedding representation to reconstruct the input sample. To the best of our knowledge, transferring knowledge from pre-trained GANs to I2I translation is not explored yet.

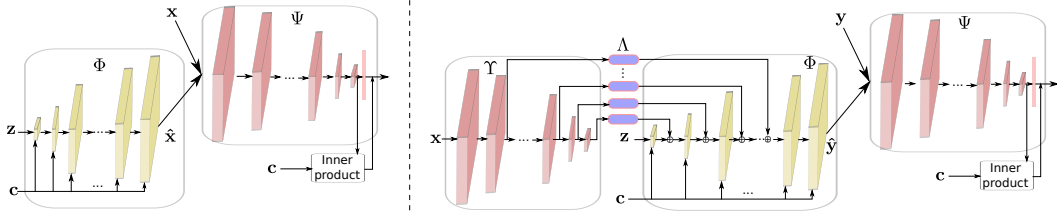

Figure 1: *Left*: the traditional form of conditional GAN (i.e., BigGAN [7]) which contains the generator $\Phi$ and the discriminator $\Psi$. *Right*: the proposed DeepI2I method based on conditional GAN (left). Our method consists of four terms: the encoder $\Upsilon$, the adaptor $\Lambda$, the generator $\Phi$ and the discriminator $\Psi$. The encoder $\Upsilon$ is initialized by pre-trained discriminator (left), as well as both the generator $\Phi$ and the discriminator $\Psi$ by pre-trained GANs (left). The adaptor $\Lambda$ aims to align the pre-trained encoder $\Upsilon$ and the pre-trained generator $\Psi$.

## 3 Proposed Approach: DeepI2I

### 3.1 Deep Hierarchical Image-to-Image Translation

We hypothesise that current I2I translations are limited due to their high-resolution bottleneck architectures. As a consequence, these methods cannot handle translations between classes with large shape changes. Low-level information, extracted in the initial layers, is not adequate for these translations, and high-level (semantic) information is required for successful translation to the target domain. The bottleneck resolution of architectures can be trivially decreased by adding additional down-sampling layers to the architectures. However, inverting a deep low-resolution bottleneck (latent) representation into a high-fidelity image is challenging [39], since the deep features contain mainly attribute-level information, from which it is difficult to reconstruct realistic images which we expect still closely follow the image structure of the input image [58, 14].

Instead, we propose a deep hierarchical I2I translation framework that fuses representations at various levels of abstraction (see Fig. 1 right). The method has a BigGAN architecture as its core, and adapts it to I2I translation by adding an encoder network which follows the same architecture as the BigGAN discriminator. This introduces several novelties to the I2I domain, such as *orthogonal regularization* to fine control over the trade-off between image quality and variety, and *class-conditioning* via a class embedding representation which operates at various levels of depth in the network. The latter allows to apply the network to many-class I2I translations. Current architectures [9, 74] which concatenate a one-hot label vector to the source image or latent space are not scalable to many-class domains.

**Method Overview.** Let $\mathcal{X}, \mathcal{Y} = \mathbb{R}^{H \times W \times 3}$ be the source and target domains. As illustrated in Figure 1 (right), our framework is composed of four neural networks: encoder $\Upsilon$, adaptor $\Lambda$, generator $\Phi$ and discriminator $\Psi$. We aim to learn a network to map the input source image $\mathbf{x} \in \mathcal{X}$ into a target domain image $\hat{\mathbf{y}} \in \mathcal{Y}$ conditioned on the target domain label $\mathbf{c} \in \{1, \ldots, C\}$ and a random noise vector $\mathbf{z} \in \mathbb{R}^{\mathbf{Z}}$, $\Phi\left(\Lambda\left(\Upsilon\left(\mathbf{x}\right)\right), \mathbf{c}, \mathbf{z}\right) \to \hat{\mathbf{y}} \in \mathcal{Y}$. We use the latent representation from different layers of encoder $\Upsilon$ to extract structural information (shallow layers) and semantic information (deep layers). Let $\Upsilon_l\left(x\right)$ be the $l$-th ($l = m, ..., n(n > m)$) ResBlock [2] output of the encoder, which is fed into the corresponding adaptor $\Lambda_l$, from which it continues as input to the corresponding layer of the generator.

As illustrated in Figure 1 (right), we take the input image $\mathbf{x}$ as input, and extract the hierarchical representation $\Upsilon\left(\mathbf{x}\right) = \{\Upsilon\left(\mathbf{x}\right)_l\}$ of input image $\mathbf{x}$. The adaptor $\Lambda$ then takes the output of $\Upsilon$ as input, that is $\Lambda\left(\Upsilon\left(\mathbf{x}\right)\right) = \{\Lambda_l\}$, where $(\Lambda_l)$ is the output of each adaptor $\Lambda_l$ which is further summed to the activations of the corresponding layer of the generator $\Phi$. We clarify the functioning of the adaptor in the next section on transfer learning, where it is used to align pre-trained representations. In case, we train DeepI2I from scratch the adaptor could be the identity function. The generator takes as input the output of adaptor $\Lambda\left(\Upsilon\left(\mathbf{x}\right)\right)$, the random noise $\mathbf{z}$ and the target label $\mathbf{c}$. The generator $\Phi$ outputs a $\hat{\mathbf{y}} = \Phi\left(\Lambda\left(\Upsilon\left(\mathbf{x}\right)\right), \mathbf{z}, \mathbf{c}\right)$ which is supposed to mimic the distribution of the target domain images with label $\mathbf{c}$. Sampling different $\mathbf{z}$ leads to diverse output results $\hat{\mathbf{y}}$.

The function of the discriminator $\Psi$ is threefold. The first one is to distinguish real target images from generated images. The second one is to guide the generator $\Phi$ to synthesize images which belong to

the class indicated by **c**. The last one is to compute the reconstruction loss, which aims to preserve a similar pose in both input source image **x** and the output $\Phi\left(\Lambda\left(\Upsilon\left(\mathbf{x}\right)\right),\mathbf{z},\mathbf{c}\right)$. Inspired by recent work [38], we employ the discriminator as a feature extractor from which the reconstruction loss is computed. The reconstruction is based on the $l$-th ResBlock of discriminator $\Psi$, and referred to by $\{\Psi_l\left(y\right)\}$.

**Training Losses.** The overall loss is a multi-task objective comprising of: (a) a *conditional adversarial loss* which plays two roles. On the one hand, it optimizes the adversarial game between generator and discriminator, i.e., $\Psi$ seeks to maximize while $\{\Upsilon,\Lambda,\Phi\}$ seeks to minimize it. On the other hand, it encourages $\{\Upsilon,\Lambda,\Phi\}$ to generate class-specific images which correspondent to label **c**. (b) a *reconstruction loss* guarantees that the synthesized image $\hat{\mathbf{y}}=\Phi\left(\Lambda\left(\Upsilon\left(\mathbf{x}\right)\right),\mathbf{z},\mathbf{c}\right)$ preserve the same pose as the input image **x**.

***Conditional adversarial loss.*** We employ GAN [18] to optimize this problem, which is as following:

$$\mathcal{L}_{adv}=\mathbb{E}_{y\sim\mathcal{Y}}\left[\log\Psi\left(\mathbf{y},\mathbf{c}\right)\right]+\mathbb{E}_{\hat{\mathbf{x}}\sim\mathcal{X},\mathbf{z}\sim p(\mathbf{z}),\mathbf{c}\sim p(\mathbf{c})}\left[\log(1-\Psi\left(\Phi\left(\Lambda\left(\Upsilon\left(\mathbf{x}\right)\right),\mathbf{z},\mathbf{c}\right),\mathbf{c}\right)\right], \quad (1)$$

where $\mathbf{p}\left(\mathbf{z}\right)$ follows the normal distribution , and $\mathbf{p}\left(\mathbf{c}\right)$ is the domain label distribution. In this paper, we leverage a conditional GAN [43]. Note that this conditional GAN [43] contains a trainable class embedding layer, we fuse it into the $\Phi$ and $\Psi$ networks. The final loss function is optimized by the mini-max game according to

$$\{\Upsilon,\Lambda,\Phi,\Psi\}=\arg\min_{\Upsilon,\Lambda,\Phi}\max_{\Psi}\mathcal{L}_{adv}. \quad (2)$$

***Reconstruction loss.*** Inspired by the results for photo-realistic image generation [26, 25, 68, 57], we consider the reconstruction loss based on a set of the activations extracted from multiple layers of the discriminator $\Psi$. We define the loss as:

$$\mathcal{L}_{rec}=\sum_{l}\alpha_l\left\|\Psi\left(\mathbf{x}\right)-\Psi\left(\hat{\mathbf{y}}\right)\right\|_1 \quad (3)$$

where parameters $\alpha_l$ are scalars which balance the terms, are 0.1 except for $\alpha_3=0.01$. Note that this loss is only used to update the encoder $\Upsilon$, adaptor $\Lambda$, and generator $\Phi$.

***Full Objective.*** The full objective function of our model is:

$$\min_{\Upsilon,\Lambda,\Phi}\max_{\Psi}\lambda_{adv}\mathcal{L}_{adv}+\lambda_{rec}\mathcal{L}_{rec} \quad (4)$$

where both $\lambda_{adv}$ and $\lambda_{rec}$ are hyper-parameters that balance the importance of each terms.

## 3.2 Knowledge transfer

Transfer learning has contributed to the wide application of deep classification networks [13]. It allows exploiting the knowledge from high-quality pre-trained networks when training networks on datasets with relatively little labeled data. Given the success of knowledge transfer for classification networks, it might be surprising that this technique has not been applied to I2I problems. However, it is unclear what dataset could function as the universal I2I dataset (like imageNet [11] for classification networks). Training an I2I network on imageNet is unsound since many translations would be meaningless (translating 'cars' to 'houses', or 'red' to 'crowd'). Instead, we argue that the knowledge required by the encoder, generator, and discriminator of the I2I system can be transferred from high-quality pre-trained GANs.

We propose to initialize the DeepI2I network from a pre-trained high-quality BigGAN (Figure 1 left). We leverage the pre-trained discriminator $\Psi$ to initialize the discriminator $\Psi$ of the proposed model, and we use the pre-trained generator $\Phi$ to initialize the deepI2I generator $\Phi$. This would leave the encoder $\Upsilon$ still uninitialized. However, we propose to use the pre-trained discriminator $\Psi$ to also initialize the encoder $\Upsilon$. This makes sense, since the discriminator of the BigGAN has the ability to correctly classify the input images on imageNet [11], which optimizes it to be an effective feature extractor.

Our architecture introduces connections between the various layers of the encoder and the generator. This allows transferring representations of various resolutions (and abstraction) between the encoder $\Upsilon$ and the generator $\Phi$. The combination is done with a dedicated *adaptor network*, and combined with a summation:

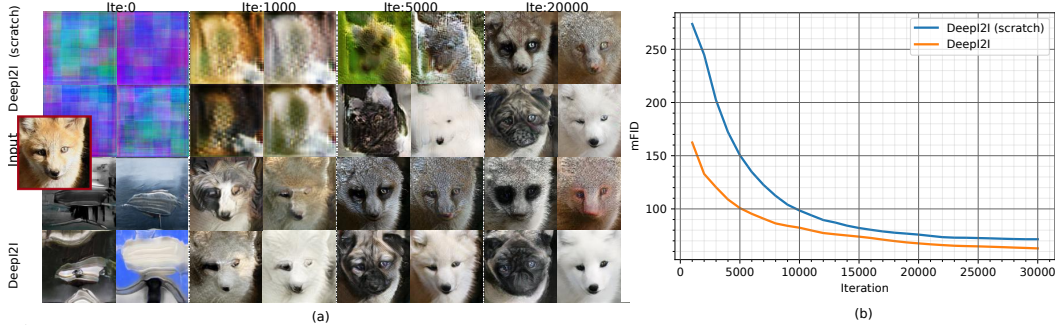

(a)                                    (b)

Figure 2: (a) Visualization of DeepI2I(scratch) in first two rows and DeepI2I in 3rd and 4th row. (b) Evolution of evaluation metrics when trained from scratch or using a pre-trained model for our method measured with mFID (source: imageNet, target: Animal faces). The curves are smoothed for easier visualization. Target animal label: meerkat, mongoose, pug and samoyed.

| Variants | RC↑ | FC↑ | mKID ×100↓ | mFID ↓ |
|---|---|---|---|---|
| DeepI2I (w-3) | 2.80 | 15.0 | 12.2 | 154.4 |
| DeepI2I (w-4) | 2.47 | 1.58 | 12.7 | 150.8 |
| DeepI2I (w-5) | 11.4 | 28.4 | 14.4 | 156.4 |
| DeepI2I (w-6) | 11.0 | 21.7 | 20.7 | 198.5 |
| †DeepI2I | 33.0 | 42.5 | 7.41 | 92.0 |
| DeepI2I | **49.5** | **55.4** | **4.93** | **68.4** |

Figure 3: Ablation study of the variants of our method on Animal faces. *Left*: the first column is the input image, following the translated outputs under variants of DeepI2I. *Right*: the correspondingly quantitative result. † means we train DeepI2I without adaptor. DeepI2I (w-3) refers to model with $\Lambda_3$ and without $\Lambda_l$ ($l = 4, 5, 6$)

$$\hat{\Phi}_l = \Phi_l + w_l \Lambda_l \tag{5}$$

where $\Phi_l$ is the output of the corresponding layer which has same resolution to $\Lambda_l$. The hyper-parameters $w_l$ are used to balance the two terms (in this work we set $w_l$ is 0.1). When we use the pre-trained encoder and decoder the representations are not aligned, i.e., the feature layer corresponding to the recognition of *eyes* in the encoder is not the same feature layer which will encourages the generator to generate *eyes*, since there is no connection between both when training the BigGAN. The task of the adaptor network is to align the two representations. The adaptors are multi-layer CNNs (see Suppl. Mat. Sec. A for details).

We train deepI2I in two phases. In the first phase, we only train the adaptors and discriminator, keeping the encoder and generator fixed. In the second phase we train the whole system except for the encoder, which we keep fixed. We empirically found that a frozen encoder results in slightly better performance.

# 4 Experiments

In this section, we will first evaluate our model on the more challenging setting, *many-class I2I translation*, which is rarely explored in the previous works. Next, we perform *two-class I2I translation* which allows us to compare to unconditional I2I methods [8, 21, 32, 34, 37, 76].

## 4.1 Experiment setting

The training details for all models are in included Suppl. Mat. Sec. A. Throughout this section, we will use *deepI2I* to refer to our method when using the initialization from BigGAN, and we use *deepI2I (scratch)* when we train the network from scratch.

**Evaluation metrics.** We report results on four evaluation metrics. Fréchet Inception Distance (FID) [20] measures the similarity between two sets in the embedding space given by the features of a convolutional neural network. Kernel Inception Distance (KID) [6] calculates the squared maximum mean discrepancy to indicate the visual similarity between real and synthesized images. We calculate the mean values of all categories in terms of FID and KID, denoted as *mFID* and *mKID*. We further evaluate on two metrics of translation accuracy to show the ability of the learned model to synthesizing the correct class-specific images, called real classifier (*RC*) and fake classifier (*FC*) [55]

| Datasets | Animal faces (*710/per class*) | | | | Birds (*78/per class*) | | | | Foods (*110/per class*) | | | |
|---|---|---|---|---|---|---|---|---|---|---|---|---|
| Method | RC↑ | FC↑ | mKID×100↓ | mFID↓ | RC↑ | FC↑ | mKID×100↓ | mFID↓ | RC↑ | FC↑ | mKID×100↓ | mFID↓ |
| StarGAN | 33.4 | 38.2 | 15.6 | 157.7 | 9.61 | 10.2 | 21.4 | 214.6 | 10.7 | 12.1 | 20.9 | 210.7 |
| SDIT | 32.9 | 39.1 | 15.3 | 151.8 | 8.90 | 8.71 | 22.7 | 223.5 | 11.9 | 11.8 | 23.7 | 236.2 |
| DMIT | 36.7 | 42.1 | 14.8 | 146.7 | 12.9 | 11.4 | 23.5 | 230.4 | 8.30 | 10.4 | 19.5 | 201.4 |
| DeepI2I (scratch) | 49.2 | 52.4 | 5.78 | 80.7 | 3.24 | 5.84 | 30.5 | 301.7 | 5.83 | 4.67 | 26.5 | 278.2 |
| DeepI2I | **49.5** | **55.4** | **4.93** | **68.4** | **20.8** | **22.5** | **8.92** | **146.3** | **30.2** | **19.3** | **6.38** | **130.8** |

Table 1: Comparison with baselines. DeepI2I obtains superior results on Animal Faces. For the datasets with less labelled data (birds and foods), DeepI2I (scratch) does not obtain satisfactory results. However, when combined with transfer learning DeepI2I significantly outperforms existing methods.

respectively. The former is trained on real data and evaluated on the generated data, while the latter is trained on the generated samples and evaluated on the real data.

**Datasets.** We present our results on four datasets, namely *Animal faces* [38], *Birds* [63], *Foods* [31] and *cat2dog* [34]. *Animal faces* dataset contains 117,574 images and 149 classes in total, *Birds* has 48,527 images and 555 classes in total, *Foods* consists of 31,395 images and 256 classes in total, and *cat2dog* composes of 2235 images and 2 classes in total. We resized all images to $128 \times 128$, and split each data into training set (90 %) and test set (10 %).

**The baselines for many-class I2I.** We compare to StarGAN [9], SDIT [67] and DMIT [74], all of which performs image-to-image translation between many-class domains. *StarGAN [9]* performs scalable image translation for all classes by inputting the label to the generator. *SDIT [67]* obtains scability and diversity in a single model by conditioning on the class label and random noise. *DMIT [74]* concatenates the class label and the random noise, and maps them into the latent representation. All these architectures are based on high-resolution bottleneck architectures.

**The baselines for two-class I2I.** We also adopt six baselines for this setting. *CycleGAN [76]* used a cycle consistency loss to reconstruct the input image, avoiding the requirement of paired data. *UNIT [37]* presented an unsupervised I2I translation method under the shared-latent space assumption. Both *MUNIT [21]* and *DRIT [34]* disentangled the latent distribution into the content space which is shared between two domains, and the style space which is domain-specific and aligned with a Gaussian distribution. *NICEGAN [8]* investigated sharing weights of the encoder and discriminator. *UGATIT [32]* studied an attention module and a new learnable normalization function, which is able to handle the geometric changes.

### 4.2   Many-class image-to-image translation

**Ablation study**   We conduct ablation studies to isolate the validity of the key components of our method: transfer learning technique and the adaptor.

*Transfer learning.* Figure 2 shows the visualization (Figure 2 (a)) and the evolution of mFID (Figure 2 (b)) during the training process without and with knowledge transfer. As shown in Figure 2 (a), the proposed method (DeepI2I) adapted from a pre-trained model is able to synthesize the shape information of class-specific images in significantly fewer iterations than the model trained from scratch (DeepI2I (scratch)) at the early training stage. And the translated images from DeepI2I are observed to have higher quality and more authentic details than DeepI2I (scratch).Figure 2 (b) also supports the conclusion: DeepI2I converges much faster than DeepI2I (scratch) in terms of mFID, though the gap is reduced over time.

We perform an additional ablation of partial transfer learning on the Animal faces dataset in Fig. 6(right). Instead of using the pre-trained GAN to initialize all three networks of the I2I system, we consider only initializing some of the networks with the pre-trained GAN and randomly initializing the remaining ones. As can be seen, the network does only successfully converge when we initialize all three networks with the pre-trained GAN.

*Adaptor.* We explore a wide variety of configurations for our approach by applying adaptors on several layers between encoder and generator. These results are based on DeepI2I initialized with the pre-trained GAN. Here we evaluate the effect of each adaptor on these four evaluation metrics. E.g. *DeepI2I* (w-3) refers to model with the adaptor $\Lambda_3$ and without adaptors $\Lambda_l$ ($l = 4, 5, 6$). Figure 3 presents a comparison between several variants of the proposed method. Only considering the single adaptor fails to achieve satisfactory performance, indicating that lacking either the information contained in the lower layer (i.e. the third, fourth, fifth ResBlocks of encoder $\Upsilon$ ) nor the semantic information extracted in deep layers (i.e. the sixth ResBlock), the method cannot conduct effective

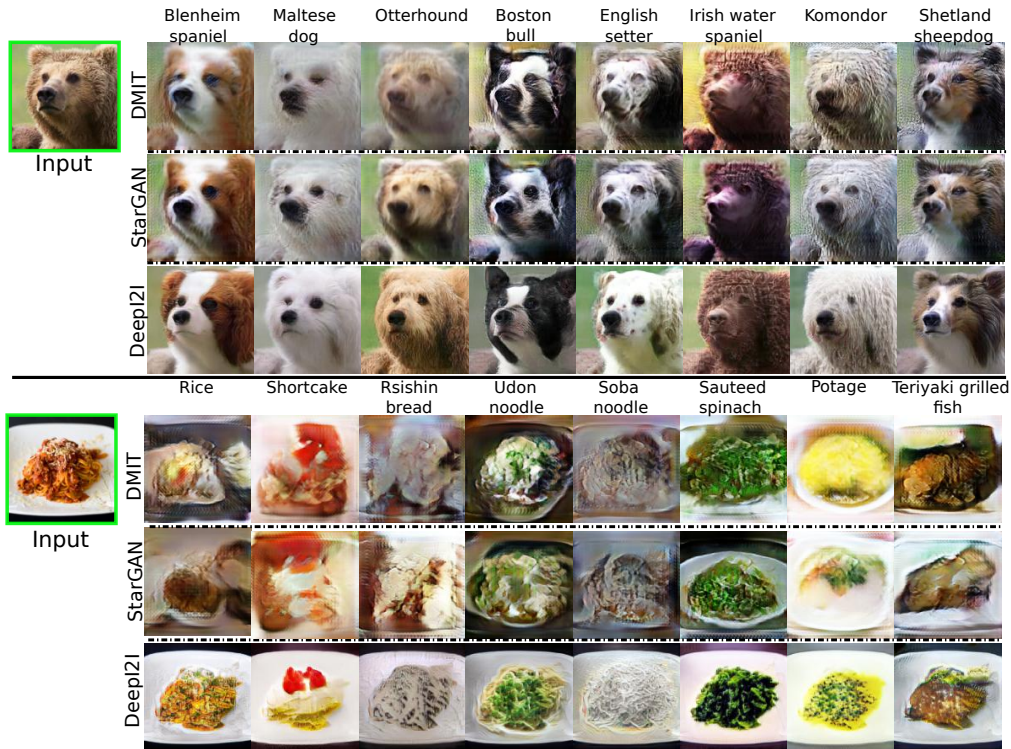

Figure 4: Qualitative comparison on animal faces and foods. The input images are in the first column and the remaining columns show the class-specific translated images. More examples in Suppl. Mat. Sec. B.

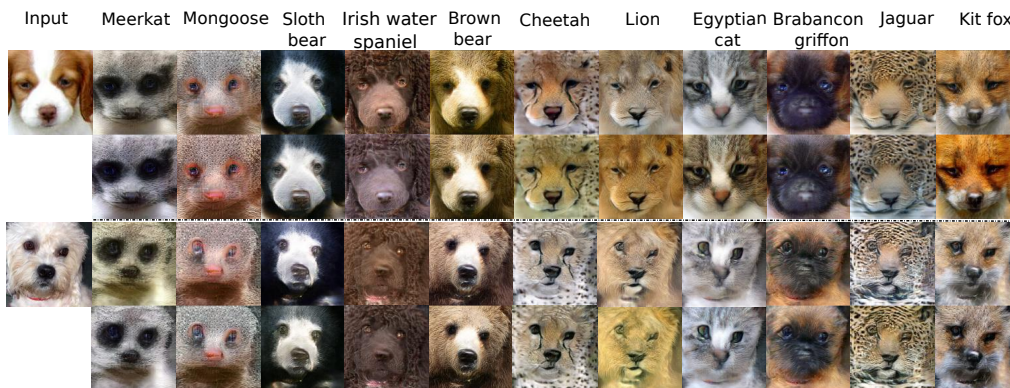

Figure 5: Qualitative results of DeepI2I. The input image is in the first column and the remaining column show the class-specific outputs. For each specific target class, we show two images.

I2I translation for challenging tasks. Notably, only using the adaptor on the deep feature (i.e. the sixth ResBlock ) obtains the worst performance, since the deep layer contains more attribute information and less structural information, which largely increases the difficulty to generate realistic image [45]. We also remove the adaptor network, and directly sum the extracted feature into the generator, called †DeepI2I. Note we connections at the all four ResBlocks. †DeepI2I obtains better results (e.g., mFID: 92.0) than the case where only one adaptor is considered. But the large gap with DeepI2I (e.g., mFID: 68.4) shows the importance of the adaptor network. The combination of adaptors (DeepI2I) in our proposed method achieves the best accuracy on all of the evaluations.

*Number of downsampling layers.* In this paper, we propose to use multiple downsampling layers to extract effectively the structure and semantic information. We perform an experiment to evaluate the effective of the number of downsampling layers. As shown in Table 6 (middle), we experimentally find that more downsampling layers results in better performance.

| Enc. | Gen. | Dis. | mFID↓ |
|---|---|---|---|
| × | × | × | 80.7 |
| × | √ | × | 276.4 |
| × | × | √ | 266.3 |
| × | √ | √ | 167.2 |
| √ | √ | √ | 68.4 |

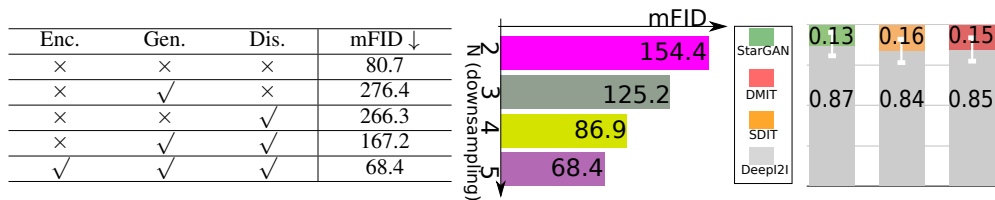

Figure 6: (left) Ablation of transfer learning, considering partial transfer of encoder (En.), generator(Gen.), and Discriminator(Dis.). (middle) Ablation on number of downsampling layers. (right) User study.

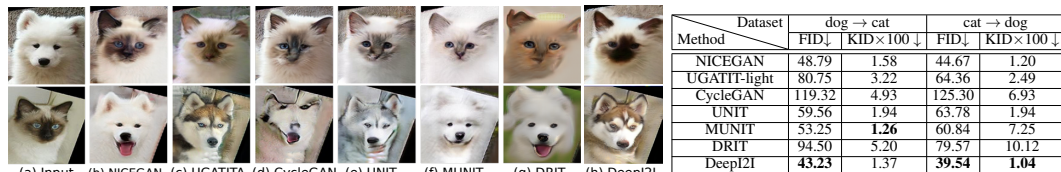

| Dataset | dog → cat | | cat → dog | |
|---|---|---|---|---|
| Method | FID↓ | KID×100↓ | FID↓ | KID×100↓ |
| NICEGAN | 48.79 | 1.58 | 44.67 | 1.20 |
| UGATIT-light | 80.75 | 3.22 | 64.36 | 2.49 |
| CycleGAN | 119.32 | 4.93 | 125.30 | 6.93 |
| UNIT | 59.56 | 1.94 | 63.78 | 1.94 |
| MUNIT | 53.25 | **1.26** | 60.84 | 7.25 |
| DRIT | 94.50 | 5.20 | 79.57 | 10.12 |
| DeepI2I | **43.23** | 1.37 | **39.54** | **1.04** |

(a) Input  (b) NICEGAN  (c) UGATITA  (d) CycleGAN  (e) UNIT  (f) MUNIT  (g) DRIT  (h) DeepI2I

Figure 7: Examples of generated outputs (left) and the metric results (right) on *cat2dog* dataset.

**Quantitative results.** Table 1 reports the quantitative results on the *Animal faces* [38], *Birds* [63] and *Foods* [31] datasets to assess many-class I2I translation. We can see that the proposed method achieves the best performance (denoted as *DeepI2I* in the table), in terms of four evaluation metrics, indicating that our model produces the most realistic and correct class-specific images among all the methods compared. In addition, we also conduct an experiment without transferring any information from BigGAN (denoted as *DeepI2I (scratch)*). On animal faces dataset, DeepI2I (scratch) obtains better results than the other baselines, and comparable performance to DeepI2I. However, DeepI2I (scratch) achieves the worst scores on birds and foods datasets. The different score of above metrics is due to the fact that the former has a large number of images per class (animal faces: 710/per class), while the latter has only limited data (birds: 78/per class; foods: 110/per class)[3]. DeepI2I wins in all metrics, clearly verifying the importance of the hierarchical architecture, combined with knowledge transfer for deep I2I translation.

Finally, we perform a psychophysical experiment with generated images by StarGAN, DMIT and DeepI2I. We conduct a user study and ask subjects to select results that are more realistic given the target label, and have the same pose as the input image. We apply pairwise comparisons (forced choice) with 26 users (30 pairs of images/user). Figure 6(right) shows that DeepI2I considerably outperforms the other methods.

**Qualitative results.** Figure 4 shows the many-class translation results on animal faces and food dataset. We observe that our method provides a higher visual quality of translation compared to the StarGAN and DMIT models. Taking the animal faces examples, both StarGAN and DMIT barely synthesize a few categories, but fail to generate realistic images for most ones. DeepI2I, however, not only manages to map the input image into class-specific image given the target label, but also guarantees to generate high-resolution realistic images. The visualization of the food dataset also supports our conclusion: DeepI2I can synthesize target images given a target food name.

**Scabality and diversity** We also explore two significant properties: the scability and the diversity. As shown in Figure 5, DeepI2I successfully translates the input image into class-specific images, clearly indicating that the proposed method provides scability. Given the target attribute (i.e., *meerkat*), our method generates diverse outputs (the second column). What is more important, we achieve both the scalability and the diversity with one single model.

### 4.3 Two-class image-to-image translation

Here we consider translating images between two categories: cats and dogs, to be able to compare our method with other unconditional I2I methods. Figure 7 (left) compares DeepI2I with six baselines. We can see how DeepI2I clearly generates more realistic target images. Figure 7 (right) reports the metric values. Our method outperforms all the baselines on the evaluation metrics except for KID

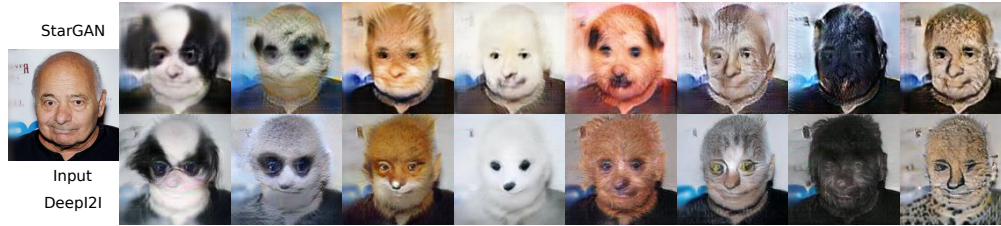

Figure 8: Unseen I2I translation: the input image is mapped into eight animal faces with StarGAN and DeepI2I.

(dog→cat). This experiment shows that our method still keeps comparable performance even for two-class image-to-image translation.

## 4.4 Unseen image-to-image translation

To illustrate the generalization power of deepI2I, we also perform unseen I2I. Here we provide the network with an input class which was not used during training. We use the network trained on the animal face and provide it with a human face. As illustrated in Figure 8, StarGAN fails to translate the input human face into the class-specific outputs, and the generated images are not satisfactory. DeepI2I, however, manages to generate high-quality images with significant shape changes (consider for example column four of results).

## 5 Conclusion

We introduced a new network for many-class image-to-image translation including classes with significant shape changes. We extract the hierarchical representation of an input image, including both structural and semantic information. Hierarchical I2I obtains inferior results on limited datasets. To mitigate this problem we propose to initialize deepI2I from a pre-trained GAN. As such we are the first to study transfer learning for I2I networks. Naively duplicating the pre-trained GAN suffers from an alignment problem. We address this by introducing a dedicated adaptor network. We qualitatively and quantitatively evaluate our method on several datasets.

## Acknowledgments and Disclosure of Funding

We acknowledge the support from Huawei Kirin Solution. We also acknowledge the project PID2019-104174GB-I00 of Ministry of Science of Spain. Our acknowledged partners funded this project.

## Broader Impact

Computer Graphics (CG) plays a key role in the creative industries, especially for the movie industry. CG is the application of computer graphics to use the computer generated graphics and animation mostly for motion pictures, television commercials, videos, printed media and video games. Our application is in the field of machine learning applied to computer graphics. More specifically, our approach can be applied for the automatic translation of faces and/or objects. Traditionally, this technique is labour intensive. Potential danger of these techniques is that they can be applied for *deepfakes* which can be used to create fake events. The I2I algorithms reflect the biases present in the dataset. Therefore special attention should be taken when applying this technique to applications where possible biases in the dataset might result in biased outcomes towards minority and/or under-represented groups in the data.

## Footnotes

[1]Current I2I methods are limited because a one-hot target label vector is directly concatenated with the source image or latent space. This technique fails to be scalable to a large number of class domains.

[2]The encoder consists of a series of ResBlock. After each ResBlock the feature resolution is half of the previous one.

[3]In the Suppl. Mat. Sec. C we also include results trained on 10% of the Animal faces datasets which confirms the importance of transfer learning for small datasets.

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
