[Supplementary Material]

# Supplementary Material
# DeepI2I: Enabling Deep Hierarchical Image-to-Image Translation by Transferring from GANs

**Yaxing Wang, Lu Yu, Joost van de Weijer**
Computer Vision Center, Universitat Autònoma de Barcelona
{yaxing, lu, joost}@cvc.uab.es

## A  Architecture and Training Details

**Model details**   The proposed method is implemented in Pytorch [11]. Our model framework is composed of four sub-networks: Encoder $\Upsilon$, Adaptor $\Lambda$, Generator $\Phi$ and Discriminator $\Psi$. We adapt the structure of BigGAN [2] to our architecture. Specifically, both the generator $\Phi$ and the discriminator $\Psi$ directly duplicate the structure of BigGAN, while the encoder $\Upsilon$ is the same as the discriminator of BigGAN except for removing the last fully connected layer. We refer readers to BigGAN to learn more detailed network information.

We connect the last four ResBlock layer representations of the encoder $\Upsilon$ to the generator, via the adaptors. The dimensions of the four different tensors are: $32 \times 32 \times 192$, $16 \times 16 \times 384$, $8 \times 8 \times 768$, $4 \times 4 \times 1536$. These four tensors are fed into four sub-adapters separately. Each of the sub-adaptor consists of one Relu, two convolutional layers (Conv) with $3 \times 3$ filter and stride of 1, and one Conv with $1 \times 1$ filter and stride of 1, except for the fourth sub-adaptor which only contains two Convs with $3 \times 3$ filter and stride of 1. We obtain four tensors as output of the adaptors: $32 \times 32 \times 384$, $16 \times 16 \times 768$, $8 \times 8 \times 1536$, $4 \times 4 \times 1536$. These are further summed to the corresponding tensors of the first four layers of generator $\Phi$.

We optimize the model using Adam [7] with batch size of 32. The learning rate of the generator is 0.0001, and the one of the encoder, adaptor and discriminator is 0.0004 with exponential decay rates of $(\beta_1, \beta_2) = (0.0, 0.999)$. We use $4 \times$Quadro RTX 6000 GPUs (27G/per) to conduct all our experiments.

**Metric Computation**   For the computation of the metrics, we randomly generate 12,800 images for each target category given the same input images, which are from the test dataset. As the classifier, we use Resent50 [4] as the backbone, followed by one fully connected layer to compute the RC and FC metrics.

## B  Additional Qualitative Results

We provide additional results for models trained on animal faces, foods, and birds in Figure 12, 13, 14.

We also show results translating an input image into all category on animal faces, foods, and birds in Figure 15, 16, 17.

## C  Results on Reduced Animal Face Dataset

We also evaluate our method using fewer animal faces. Specifically, we randomly select 10% of data per class. As reported in Table 2, training with fewer animal faces per class by DeepI2I (scratch) fails

Figure 8: Interpolation by keeping the input image fixed while interpolating between two class embeddings. The first column is the input images, while the remaining columns are the interpolated results. Top rows interpolation results from *pug* to *mongoose*. Bottom rows interpolation results from *komondor* to *brown bear*.

| Datasets | Animal faces (*710/per class*) | | | |
|---|---|---|---|---|
| Method | RC↑ | FC↑ | mKID×100↓ | mFID↓ |
| DRIT++ | 35.4 | 33.1 | 14.1 | 138.7 |
| StarGAN v2 | 39.7 | 40.7 | 6.45 | 85.8 |
| DeepI2I | 49.8 | 55.4 | 4.93 | 68.4 |

Figure 9: Further results on the Animal faces dataset.

to obtain satisfactory results (e.g., mFID: 280.3), while DeepI2I with knowledge transfer achieves superior results (e.g., mFID: 141.7), which meets our expectation that the model benefits from the transfer learning especially for small dataset. Actually, these results are comparable to does of StarGAN, SDIT and DMIT trained on all data (compare to Table 1 in main paper).

## D  Further results for Animal faces dataset

We also compare our results on the Animal faces dataset with two concurrent works: DRIT++[8], and StarGAN v2 [3]. As shown in Figure 9, we outperform these recent works for all metrics.

## E  Interpolating

Figure 8 reports interpolation by freezing the input images while interpolating the class embedding between two classes. Our model still manages to generate high quantity images even given never seen class embeddings.

Figure 10: Qualitative results on the cat2dog dataset. The generated images are prone to suffer from artifacts (the red box), which is caused by StyleGAN [5].

Figure 11: Typical failure case of our method. The first column is the input, followed by the output for different target classes in the remaining columns.

## F    Results on StyleGAN

We also explore the proposed method on StyleGAN [5]. Like the one in the main paper which transfers from BigGAN, we build our model based on the StyleGAN structure, and initialize our model using the pre-trained StyleGAN, which is optimized on HHFQ dataset [5]. We refer readers to StyleGAN paper for more details.

As illustrated in Figure 10, DeepI2I manages to translate the input image (*dog*) into target image (*cat*). The generated images, however, suffer from the artifacts (the red box in Figure 10), which is a result of both the architecture and the training method of StyleGAN. In the future, we will adapt our method to the latest methods (e.g., StyleGAN2 [6]) to avoid this problem.

## G    Ablation on Reconstruction loss

The reconstruction loss helps to maintain the structure/poss information of the input sample. We conduct experiment by removing the reconstruction loss during training, and find the structure of both input (Figure 19 first column) and output (Figure 19 third column) is different. This shows that the structure information will be lost without reconstruction loss. The same is also used in [9, 12].

## H    Impact of transfer learning on translation classic

We also performed an experiment on the classic translation from horses to zebras proposed in [13]. The experiment and visualisation of the results are shown in Figure 18. In terms of FID score the results improve from 77.2 for CycleGAN to 63.2 for DeepI2I.

## I    T-SNE

We investigate the latent space of the generated images. We randomly sample 1280 images, and translate them into a specific class (*Blenheim spaniel*). Specifically, given the generated images we firstly perform Principle Component Analysis (PCA) [1] to extracted feature, then conduct the T-SNE [10] to visualize the generated images in a two dimensional space. As illustrated in Figure 20,

| Datasets | Animal faces (*71/per class*) | | | |
| Method | RC↑ | FC↑ | mKID×100↓ | mFID↓ |
| --- | --- | --- | --- | --- |
| DeepI2I (scratch) | 6.12 | 3.24 | 28.9 | 280.3 |
| DeepI2I | **30.1** | **32.4** | **15.0** | **141.7** |

Table 2: Results on reduced animal face dataset where we randomly select 10% images per class. DeepI2I obtains significantly better performance than DeepI2I (scratch).

| Datasets | Animal faces (*710/per class*) | | | |
| Method | RC↑ | FC↑ | mKID×100↓ | mFID↓ |
| --- | --- | --- | --- | --- |
| DeepI2I (scratch) | 49.2 | 52.4 | 5.78 | 80.7 |
| *DeepI2I | 11.7 | 25.9 | 15.9 | 167.2 |
| DeepI2I | **49.5** | **55.4** | **4.93** | **68.4** |

Table 3: Ablation study of the variants of our model. * means the encoder of our model is from scratch, while both the generator and the discriminator are from the pre-trained GAN (BigGAN).

given the target class (Blenheim spaniel), DeepI2I correctly disentangles the pose information of the input classes. The T-SNE plot shows that input animals having similar pose are localized close to each other in the T-SNE plot. Furthermore, it shows DeepI2I has the ability of diversity.

We also conduct T-SNE for 14,900 generated images across 149 categories (Figure 21).

## J  Failure cases

Our method obtains compelling performance for many cases, but suffers from some failures. Figure 11 shows a few failure cases. The input images are different from the normal animal faces. E.g. a side-view of a face, which is rare in the dataset, causes the model to produce unrealistic results.

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

Figure 12: Qualitative results on the Animal faces dataset. We show four examples, each one is randomly translated into 24 categories. Input images are in green boxes.

Figure 13: Qualitative results on Birds dataset. We show four examples, each one is randomly translated into 24 categories. Input images are in green boxes.

Figure 14: Qualitative results on the Foods dataset. We show four examples, each one is randomly translated into 24 categories. Input images are in green boxes.

Figure 15: Qualitative results on the Animal faces dataset. We translate the input image (top left) into all 149 categories. Please zoom-in for details.

Figure 16: Qualitative results on the Food dataset. We translate the input image (top left) into all 256 categories. Please zoom-in for details.

Figure 17: Qualitative results on the Birds dataset. We translate the input image (top left) into all 555 categories. Please zoom-in for details.

| Input | DeepI2I | CycleGAN |
|-------|---------|----------|

Figure 18: Evaluation on horse to zebra translation.

| Input | | DeepI2I | DeepI2(w/o Recon) |
|-------|--|---------|-------------------|

Figure 19: Ablation on the reconstruction loss. Results show the importance of the reconstruction loss to maintain the image structure.

Figure 20: 2-D representation of the T-SNE for 1280 generated images, the target class is *Blenheim spaniel*. Note that for each pair image, the left is the input and the right is the output image. Please zoom-in for details.

Figure 21: 2-D representation of the T-SNE for 14900 generated images across 149 classes. Please zoom-in for details.