[Reviews · NeurIPS 2020]

Review 1

Summary and Contributions: This work focuses on the multi-domain image-to-image (I2I) translation problem. The authors propose to use the train the I2I model from the pre-trained BigGAN generator and discriminator. The author also proposes an adaptor to improve performance. Experiments are conducted on several multi-domain I2I translation tasks.

Strengths: 1. The high-level idea of this work is sound. 2. The proposed method is reasonable. 3. The proposed method demonstrates superior performance compared to the baseline in the experiments.

Weaknesses: 1. The notations, equations in the method section are not clear. In Line 110 for instance, the equation $\Upsilon(x)=\{\Upsilon(x)_l\}$ is confusing. 2. The discriminator on the left side of Figure 1 is not the network used by the existing I2I methods (e.g., BicycleGAN concatenates the one-hot vector with the image as the input.) 3. Two highly-related frameworks targeting multi-domain I2I [1,2] are not cited, discussed, and compared in the paper. 4. In the table of Figure 3, it is not clear why training with partial adaptor performs worse than that of training without the adaptor? 5. Since the model is pre-trained from the BigGAN model trained on the natural images, what is the performance of the proposed method on the I2I tasks with unnatural images (e.g., face to artistic portrait)? [1] Lee, et al. "Drit++: Diverse image-to-image translation via disentangled representations.". [2] Choi et al. "StarGAN v2: Diverse image synthesis for multiple domains."

Correctness: Yes.

Clarity: See Weakness #1.

Relation to Prior Work: See Weakness #3.

Reproducibility: Yes

Additional Feedback:


Review 2

Summary and Contributions: The paper introduces a novel architecture and training scheme for the task of Image to Image translation with two main motivations: The first is the ability to generate translations in which large geometric shape changes are required. The second is to apply knowledge transfer from pre-trained GAN models (specifically BigGAN) to allow superior quality in generated results. The paper also addresses the more challenging task of class conditioned image to image translation, in which case we are required to translate to a specific target class in the target domain. To address the first motivation, the paper proposes training a Unet like architecture with many down-sampling layers thus reaching more "semantic information" and avoid looking low level information by using adapter layers. For the second motivation, it uses leverages pretraining from BigGAN for both the encoder, generator and discriminator.

Strengths: With regards to the first motivation of the paper, the method shows in Fig.3 that using the adapter layers in the formulation is important and affects the quality of the results. In addition, both qualitative and quantitative evaluation suggest and improvement in visual quality of results compared to baseline methods. Visually, alignment seems to be improved. With regards to the second motivation, training from scratch results in inferior quality compared to starting from pretrained networks, which is demonstrated both visually and numerically in Figure 2.

Weaknesses: There are a number of weaknesses I find in the paper: The first, and to me, the most significant one, is with regards to the evaluation metrics. Although 4 are provided, none of them address the ability to correctly translate from domain A to B compared to baseline method. 2 of the metrics consider the quality and diversity of results (FID and KID) and 2 others with regards to correct class of translation in the target domain. With regards to the first motivation, when using 6 down-sampling layers, it is clear that using adapter layers is useful. However, it is not clear that using 6 down-sampling layers is at all required. This seems to be a main premise of the paper: that using many downsampling layers will allow a more semantic change and thus a significant change of shape, but this is not at all demonstrated. For example, DRIT [1] uses fewer number of downsampling layers and seems to also change the shape of images. Third, related to the first weakness, while using pretainred GANs results in better quality of generation, it is not clear whether it affects to alignment of solutions. Indeed this may be simply a result of using the BigGAN architecture, which is not novel, as the generator. Further no ablation is provided with regards to which pretrained component is important for the task: Could we start just with a pretrained generator or do all components need to be pretrained? Other points: In lines 172-175, it seems to me that the encoder is not trained at all, why is this so? [1] DRIT++: Diverse Image-to-Image Translation via Disentangled Representations. ECCV 2018.

Correctness: The evaluation provided seems correct. See comments regarding evaluation of the method above.

Clarity: The paper is relatively clear. However I find the two main motivations relatively distinct, and it is not clear to me why they are provided in a one paper, and how they relate.

Relation to Prior Work: I find that related work section has few important citations missing. Since geometry is key to this paper I find a discussion of [2] important. Also, the work of [3] and follow up works make use of pretraining with GANs (hence transfer knowledge) and it also makes the distinction between lower layers which main affect structure and higher levels that affect semantic information in the context of image to image translation. [2] TransGaGa: Geometry-Aware Unsupervised Image-to-Image Translation. CVPR 2019. [3] One-Shot Unsupervised Cross Domain Translation. NeurIPS 2017.

Reproducibility: Yes

Additional Feedback: Typo: Fig1 description. Generator symbol is phi. Post rebuttal: I thank the authors for the rebuttal, which alleviated some but not all of my concerns. Regarding evaluation, the addition of the user study in the rebuttal is very helpful as it demonstrates that alignment is improved compared to baselines (i.e pose is maintained). I still believe a comparison to [2] is important as it tackles the problem of the geometry change in I2I. Regarding the need for many downsampling layers, in the rebuttal it is shown that FID clearly improves, which indicates a better generation quality. However, once again, it is not clear that alignment is not adversely affected. The same is true with using BigGAN architecture. It is not a surprise that BiGAN, a SOTA architecture for unconditional generation, results in high generation quality. But its effect on the alignment is not clear. As some of my concerns were addressed I raise my score to 5.


Review 3

Summary and Contributions: The paper introduces a new architecture for image-to-image translation that combines features from both shallower and deeper layers. Moreover, the proposed architecture enables reusing the weights from a pretrained BigGAN. The proposed methods achieves better metrics than existing I2I methods.

Strengths: I have seen many attempts to use pretrained GAN models in image synthesis, but it seems hard to show the benefit of pretraining. The proposed method actually does show positive result of doing so, by achieving superior metrics to existing methods.

Weaknesses: I think Image2StyleGAN(Abdal et al) is relevant work. Using a pretrained network, they reconstruct the image by optimizing layerwise features. This is similar to the encoding step of the proposed method, except that an encoder and adaptor is used instead of test-time optimization. The resolution of the training is quite low. I wonder if it is because of limitation of the method that doesn't leverage high-res bottleneck as much as other models, or because the authors were limited by computational resources. There is no evaluation on how much the structure of the original image is kept. RC, FC, KID and FID all measure the quality of the output, but they don't measure the correspondence between input and output. It's more important for the submitted paper to show that the input structure is maintained, because it removes the high-res bottleneck of existing I2I architecture. Please see the other sections for more comments.

Correctness: I am not convinced about the reconstruction loss of Equation (3). I think it only makes sense to impose the reconstruction loss between input and output of the same class. If the class of hat{y} is different from x, the visual needs to change. However, since the discriminator has no incentive to be invariant to the class label, it seems impossible to meet the reconstruction loss by only optimizing the encoder, adaptor and the generator network. L150: "training an I2I network on imageNet is unsound since many translations would be meaningless. " However, you do train an I2I network using a pretrained BigGAN on ImageNet. Isn't this self-contradicting?

Clarity: The description is reasonably well-written and organized, although the figures can be polished for better presentation. It's a rather minor detail, but the text within figures are not aligned or properly displayed.

Relation to Prior Work: The paper cites fairly recent works such as UGATIT and DMIT. A possible related work is (Shrivastava et al., 2017, https://arxiv.org/pdf/1612.07828.pdf) which performs image-to-image translation without cycle consistency loss. The proposed method differs from many existing works in that the cycle loss is not used. Also, it can be mentioned that the proposed architecture resembles U-Net of pix2pix, in that it consists of skip connections at each depth of layers.

Reproducibility: Yes

Additional Feedback: Just out of curiosity, I wonder if the proposed method can outperform CycleGAN on horse2zebra dataset. I found that horse2zebra dataset is quite hard for many methods because the dataset size is small. Since the proposed method leverages pretraining, and horse2zebra belong to ImageNet, the proposed method may perform better than CycleGAN. ---------------------- Thank you for the author feedback. The user study and experiment result on horse2zebra convinced me toward more positive rating of the paper.


Review 4

Summary and Contributions: This paper proposes a novel deep hierarchical image-to-image translation method, called DeepI2I which leverages hierarchical features to handle the cases that the translations require large shape changes. Besides, for the first time, the authors introduce a transfer learning strategy using a pre-trained GAN for model initialization. Furthermore, an adaptor network is contributed to addressing the misalignment problem raised in the transfer learning process. Extensive experiments are conducted to demonstrate the effectiveness of the model.

Strengths: 1. The idea that leveraging the hierarchical features to improve translation performance is intuitive and effective. This design is especially useful when the translation involves large shape changes. 2. To the best of my knowledge, this work for the first introduces the strategy of using pre-trained GANs for initialization. Although this strategy is a common practice for the recognition task, it is novel for the I2I task. 3. The authors conduct extensive experiments to demonstrate the contributions of each design choice.

Weaknesses: 1. As one of the main contributions, transfer learning from the pre-trained GAN seems effective, e.g., using a pre-trained discriminator to initialize both the discriminator and the encoder of the proposed model. However, it is not clear how such an initialization strategy affects each network? For example, what are the results when only initializing discriminator using the pre-trained model (while the encoder is randomly initialized)? 2. The contribution of hierarchical representation for the I2I task is not well demonstrated. The comparison between the DeepI2I (scratch and w/o adaptor) and DeepI2I (scratch and with adaptor) is expected to show the hierarchical representation's effectiveness. [Comments after author feedback] The authors addressed my concerns well in the rebuttal (e.g., ablation study in transfer learning, reconstruction loss), although they mistakenly referred me to R2.3 (should be R2.4) for the first point and R3.4 (should be R3.3) for the fourth point. The proposed method is novel and is proved to be effective, so I will keep my initial score (accept, 7).

Correctness: Yes

Clarity: Yes

Relation to Prior Work: Yes. This paper illustrates the limitations of the previous methods and then introduces their approach to address these limitations.

Reproducibility: Yes

Additional Feedback: 1. Could you provide more insights about the choice of w_l, i.e., how does w_l affect the performance? 2. What are the effects of the reconstruction loss? Why does this loss necessary?

[Author Response · NeurIPS 2020]



Figure 19: (left) Comparison with StarGAN v2, DRIT++, and ablation of reconstruction loss, (middle) evaluation on human face to portrait, (right) evaluation on horse to zebra translation (zoom for close inspection).

| Datasets | Animal faces (*710/per class*) | | | | Enc. | Gen. | Dis. | mFID ↓ |
|---|---|---|---|---|---|---|---|---|
| Method | RC↑ | FC↑ | mKID×100↓ | mFID↓ | × | × | × | 80.7 |
| DRIT++ | 35.4 | 33.1 | 14.1 | 138.7 | × | √ | × | 276.4 |
| StarGAN v2 | 39.7 | 40.7 | 6.45 | 85.8 | × | × | √ | 266.3 |
| DeepI2I | 49.8 | 55.4 | 4.93 | 68.4 | √ | √ | √ | 68.4 |

Figure 20: (a)User study, (b)study of the number downsampling, (c)related frameworks, (d)ablation of transfer learning. Note En.: encoder, Gen.: generator, Dis: discriminator

We thank the reviewers for their feedback: the paper proposes a *sound* (R1), *novel* (R4) method
with *novel/new architecture* (R2/R3), obtaining *superior results* (R1, R3). It is the *first paper using*
*pre-trained GANs for I2I initialization* (R4). We will comment the many requested experiments more
completely in any final version. We will improve related work with mentioned papers.

**R1** **R1.1 Discriminator I2I methods:** Figure 1(left) depicts a generative model (loosely based on
BigGAN) which has not been applied to I2I before. We show that our target-label conditioning is
more scalable (see also lines 98-99 in the main paper), while conditioning by one-hot vector does not
scale well to many-class I2I. **R1.2 Related frameworks:** We report the result of both *DRIT++* and
*StarGAN v2* in Figs. 19(left) and 20(c). We outperform them on all 4 metrics. Note that both these
methods do not address transfer learning for I2I. **R1.3 Results of adaptor:** The *without adaptor*
setting does have skip connections for layers 3-6 but does not have adaptor layers. The partial adaptor
setting only considers a single connection. This shows that the hierarchical connections are crucial
for good performance. **R1.4 Human face to portrait[29]:** we show the generated images (Fig. 19
(columns 6-9)), and obtain the FID/KID for these methods: DeepI2I/StarGAN/DMIT: (160.3/8.8) /
(189.4/9.7) / (194/9.6), indicating that DeepI2I has a slight advantage.

**R2** **R2.1 Evaluation metrics:** We conduct a user study and ask subjects to select results that are
*more realistic given the target label, and have the same pose as the input image*. We apply pairwise
comparisons (forced choice) with 26 users (30 pairs image/user). Fig. 20(a) shows that DeepI2I
considerably outperforms the other methods. **R2.2 Downsampling layers:** In Fig. 20(b) we explore
the downsampling layers: more downsampling layers results in better performance. **R2.3 Relation**
**of contributions:** BigGAN-like architectures have not been explored for I2I (contr. 1). We argue and
show that this helps to scale to many-class I2I problems. Directly training such architectures results
in unsatisfactory results for small domains (see Tables 1 and Suppl. Mat. Sec. C). However, when
combined with a pre-trained GAN (contr. 2), we obtain state-of-the-art results for many-class I2I
problems, even those with fewer images per class. **R2.4 Ablation transfer learning:** We performed
additional ablation of partial transfer learning in Fig. 20(d) (see also Suppl. Mat. Sec. F). In case of
partial transfer networks suffer from mode collapse leading to unsatisfactory results.

**R3** **R3.1 Low resolution:** we also trained our model on styleGAN with image size 256*256, and
get high quality results (see Suppl. Mat. Sec. E). **R3.2 Translation of structure:** We agree with
the reviewer that it would be nice to be able to measure the success of the structure translation.
However, currently no evaluation metrics exist. Therefore, we propose a user study (see **R2.1**). **R3.3**
**Reconstruction loss:** we removed the reconstruction loss during training, and find the structure of
both input (Fig. 19 first column) and output (Fig. 19 fifth column) is different, indicting that the
structure information will be lost without reconstruction loss. The same loss also appears in [32][55].
**R3.4 Self-contradiction in transfer learning motivation:** The point we wanted to make here is
subtle. It is not clear how to train a universal I2I network on ImageNet, since many classes cannot
be translated to each other. Instead, we show the pretrained GAN features are useful for I2I. **R3.5**
**Horse2zebra** We performed the experiment and visualize results in Fig. 19 (right). We obtain a better
FID score, i.e., (DeepI2I:CycleGAN): (63.2:77.2).

**R4** **R4.1 Ablation transfer learning:** We refer to **R2.3**. **R4.2 DeepI2I (scratch and w(w/o) adap-**
**tor):** we report result in this setting, and experimentally find that the mFID is 186.4 without adaptor,
and less than DeepI2I (80.7 in Tab.1) . **R4.3 Choice of** $w_l$**:** We normalize features of both $\Phi_l$ and $\Lambda_l$
and used $w_l$=0.1 in all experiments. We found this to slightly faster convergence than $w_l$=1. We did
not experiment with other settings. Ideally, this parameter would be optimized with cross-validation.
**R4.4 Reconstruction loss:** The reconstruction loss helps align the two domains. We refer to **R3.4**.



[Meta-Review · NeurIPS 2020]

Reviewers were split on this paper, with three recommended accept and one recommending rejection. The reviewers appreciated the success at getting pretraining to work for GANs, which has been an open problem. However, reviewers also had concerns about the thoroughness of the experiments, especially that structure preservation was not evaluated in the submission (R2, R3). The rebuttal did a good job addressing this concern, with a user study, although in discussion the reviewers requested a study with more participants for the final version of the paper. This and other results provided in the rebuttal convinced most reviewers and the AC of the merits of this paper. These additional results should definitely be added to the final version of the paper. Reviewers also pointed out several missing references that should at least be discussed, and ideally compared against (in particular, see R2's comments about TransGaGa). With these changes the submission could become a substantially stronger camera ready.